# Multi-Level Optimization and Strategies in Microbial Biotransformation of Nature Products

**DOI:** 10.3390/molecules28062619

**Published:** 2023-03-14

**Authors:** Dan Qin, Jinyan Dong

**Affiliations:** 1Key Laboratory of Eco-environments in Three Gorges Reservoir Region (Ministry of Education), School of Life Sciences, Southwest University, Chongqing 400715, China; 2Key Scientific Research Base of Pest and Mold Control of Heritage Collection (State Administration of Cultural Heritage), Chongqing China Three Gorges Museum, Chongqing 400060, China; 3Chongqing Key Laboratory of Optical Fiber Sensor and Photoelectric Detection, Chongqing Engineering Research Center of Intelligent Optical Fiber Sensing Technology, Chongqing University of Technology, Chongqing 400054, China

**Keywords:** natural products, microbial transformation, fermentation, multi-level optimization, synthetic biology

## Abstract

Continuously growing demand for natural products with pharmacological activities has promoted the development of microbial transformation techniques, thereby facilitating the efficient production of natural products and the mining of new active compounds. Furthermore, due to the shortcomings and defects of microbial transformation, it is an important scientific issue of social and economic value to improve and optimize microbial transformation technology in increasing the yield and activity of transformed products. In this review, the aspects regarding the optimization of fermentation and the cross-disciplinary strategy, leading to the microbial transformation of increased levels of the high-efficiency process from natural products of a plant or microbial origin, were discussed. Additionally, due to the increasing craving for targeted and efficient methods for detecting transformed metabolites, analytical methods based on multiomics were also discussed. Such strategies can be well exploited and applied to the production of more efficient and more natural products from microbial resources.

## 1. Introduction

Natural products are organic compounds produced by biological systems, which can be divided into two categories: primary compounds and secondary metabolites. Among them, secondary metabolites have attracted much attention because of their multiple biological effects on other organisms. In the past few decades, the market of botanical drugs has increased significantly, in addition to the use of spices, pigments, essential oils, antioxidants, etc. Plenty of secondary metabolites derived from medicinal plants are being used currently in medicines, which are, in most of the cases, complementary to drug use [1].

Microorganisms can produce a large number of rich enzymes under specific nutritional conditions and can produce various products that are difficult to produce by traditional synthesis methods [2]. In recent years, as an important manufacturing tool, microbial transformation has gained more and more recognition in the pharmaceutical industry. Biocatalysis can simplify the synthesis of some drugs or intermediates. Compared with chemical synthesis, they eliminate the complex separation and purification steps and selectively produce the target product under relatively mild fermentation conditions [3]. Additionally, the process of microbial transformation operates in a near-neutral pH, temperature, and atmospheric pressure, while chemical synthesis usually requires extreme conditions. Microbial transformation also has high response specificity, enantioselectivity, and regional specificity [4] (Figure 1).

In order to obtain microbial transformation technologies with industrialized value, many researchers have carried out several years of research on the selection of microorganism transformation strains and optimization of transformation conditions and systems [1,2,3]. Biotransformation is problematic because of the toxicity and inhibition of reactants and products. If the substrate is toxic, it can kill microorganisms and prevent any transformation. Due to the limitation of time and the lack of transformation strains, microbial transformation has become the second choice of the pharmaceutical industry. Therefore, it is necessary to develop more transformation potential of microbial strains in order to convert various substances into the products needed, especially the active pharmaceutical ingredients. Owing to the complex biological system involved, the yield of microbial transformation is much lower than that of chemical synthesis [2,5]. Furthermore, due to their complex structure, some bioactive compounds are only obtained from natural plants. One improved approach is represented by the genetic engineering and enzyme engineering techniques, leading to the increasing production of biotransformation [4]. The biotransformation of transgenic microorganisms and their ability to release products provide an alternative method for natural products with low contents in nature. The strain with the enzyme gene can produce abundant soluble and functional enzymes to convert the substrate into the target product. This method is likely to become an alternative method for traditional separation methods [6,7]. However, there are many problems in the biotransformation of microorganisms: (1) the growth burden of host cells is caused by exogenous enzyme proteins; (2) the metabolic pathway of host cells is disturbed by the reconstructed metabolic pathways; (3) the accumulation of toxic and side-effect products [8] (Figure 1).

Another effective approach to improve the productivity of microbial transformation is the optimization of the culture medium and the selection of host strains [6]. Many factors affect microbial transformation, such as fermentation mode, culture medium type, fermentation temperature, humidity, time, pH, rotation speed, substrate solubility, and co-factors [7,9,10,11,12,13,14]. Moreover, due to the existence of plant enzyme systems and the differences of oxygen content, the products of solid-state fermentation and liquid fermentation are quite different [9,15,16,17]. For example, microorganisms secrete different enzymes at different times in the biotransformation process, which makes the characteristic metabolites have characteristic generation times. In the growth process of microorganisms, the production of metabolites changes the pH of the medium, which makes the addition time of different substrates change the pH, thus affecting the yield of the transformed products [14]. Fermentation media can affect the output of conversion products by affecting cell growth and enzyme activity, while co-factors such as metal ions can change the pH of the media and use them as a catalyst for reaction, and both microbial growth and enzyme activity require an optimum temperature to achieve maximum yield transformation. Additionally, surfactant is similar to phospholipid molecules in cell membranes in the structure, which can improve the conversion efficiency by improving the permeability of the cell membrane [7,12,13]. Specially, endophytes often depend on the host’s microbial transformation and have the ability of transformation, while the transformation substrate is the only carbon source or nitrogen source to screen strains with the ability of transformation in the natural environment [13,14,15] (Figure 1).

The tolerance of host cells to toxic products is the bottleneck of microbial transformation, which is more significant in the early stage of microbial transformation. At the same time, some secondary metabolites are difficult to develop and use due to their low content, but they are often important sources of microbial drugs. Through the method to change metabolic flow based on synthetic biology, the synthesis period of target products can be dynamic in regulation. Combined with genome mining and analysis of metabonomics, the metabolic flow of bacteria can be changed, and more carbon flow in the cell can flow to the synthesis of secondary metabolites [8,18]. However, it is still difficult to synthesize some complex compounds directly through microorganisms. The combination of microbial synthesis and organic synthesis provides an opportunity for natural products that are difficult to synthesize directly [19]. Microorganisms can carry out various required biochemical transformations, but their efficiency is low, and their stability is low. Therefore, it is extremely important to design the metabolic process of high-efficiency cell factories. One of the development trends of microbial transformation is to optimize the process of industrial strains by using transcriptome, proteome, metabonomics, and their combinations [20] (Figure 1).

In the past 10 years, our research group has carried out a lot of research work on the biotransformation of plant natural products by endophytic fungi, and we have also made a series of research progress [15,21,22,23,24,25,26,27]. In order to make this progress better and provide references to relevant studies, we carried out this review. This review covers the optimization strategies of microbial biotransformation, including selecting potential strains, optimizing culture and fermentation conditions, synthetic biology strategies, combination chemical synthesis and biosynthesis, multi-strains co-culture strategy, and multi-omics strategies, and 124 references from 1999 to 2022 are cited. In addition, literature review in relevant fields shows that this is the latest and most comprehensive and systematic review of multi-level optimizations and strategies of microbial biotransformation of natural products. In the past, reviews of microbial transformation mainly focused on the presentation of a certain type of special substances, such as sterols, flavonoids, saponins, triterpenes, and so on, while ignoring the multiple strategies and optimization of biological biotransformation. Therefore, such a systematic summary has important references and guiding significances for the research of the microbial biotransformations of natural products. 

## 2. Select Microbial Strains for Biotransformation

### 2.1. Isolation of Transformed Microbial Strains

Endophytes are widespread in plants, and some are beneficial to host plants. Considering that endophyte bacteria have a long-term interaction with the host during the evolutionary adaptation process, endophytes have special transformation potential for the metabolites of host plants [15]. For instance, the endophytic fungus *Ceriporia lacerate* hs-zjut-c13a from *Huperzia serrata* transformed huperzia into five compounds and three trees sesquiterpenoids, in which huptremules A-D have the characteristics of functional metals (trees sesquiterpenoids) and the exogenous substrata (HupA) [28]. Similarly, its endophytic fungus *I. lacteus* transformed huperzine A into 8*α*,15*α*-epoxyhuperzine A [29]. Another endophytic fungus *Bjerkandera adusta* transformed huperzine B into 8α,15*α*-epoxyhuperzine B, hupB analogs, 16-hydroxyhuperzine B, and carinatumin B [30]. Moreover, the endophytic fungus *Fusarium oxysporum* of *Catharanthus roseus* transformed vinblastine (*Catharanthus roseus*) into vincristine [31]. The endophytic *Xylaria* sp. isolated from *Chinchona pubescens* transformed cinchona alkaloids (*Chinchona pubescens*) into 1-N-oxide derivatives [32]. The ginsenosides Rg1, Rh1, Rb1, and Re are the major sapoinsin in *Panax notoginseng*, which is transformed into a new compound and nine known compounds by the endophytic fungi anax notoginseng genera *Fusarium*, *Nodulisporium*, *Brevundimonas,* and endophytic bacteria *Bacillus* genera of *Panax notoginseng* [33]. In addition to that, the solid-state fermentation of the host plant can also be carried out by endophytes. For example, by an endophytic fungus *Umbelopsis dimorpha* grown on host-plant *Kadsura angustifolia* and wheat bran, a variety of new compounds including triterpenes, sesquiterpenes, and monoterpenes have been produced [15]. Fermentation of *K. angustifolia* with endophytic fungus *Penicillium* sp. induced production of highly oxygenated schitriterpenoids from nigranoic acid, including nine new triterpenes, four of which are rare high-oxidation triterpenes [21]. Five new triterpenes with high-oxidation degrees were isolated by secondary fermentation of the fungi *Penicillium* sp. [22]. The entomogenous fungus *Paecilomyces gunnii* transformed caffeine (Oolong tea) into theophylline and 1,7-dimethylxanthine [34]. These examples proved that endophytes have great potential in the fermentation and transformation of host plants.

Compared with plant endophytes, microbial species of natural environments are more abundant and diverse. *Mucor circinelloides lusitanicus* isolated from soil can introduce a 7*α*-hydroxyl group into 5-en-3β-ol steroids [35]. For instance, *Trichoderma harzianum* isolated from soil could convert steriodal and saponins (*Dioscorea zingiberensis* C.H.Wright) into diosgenin [12]. The steroidal saponins (*Agave sisalana Perrin*) were transformed into tigogenin by *Bacillus subtilis* zg-21 isolated from the soil in a karst area of Guilin [36]. Ginsenosides Rb1, Rb2, and RB3 (*Panax notoginseng* leaves) were transformed into ginsenoside compound K by *Paecolomies bainier* sp.229 isolated from the soil samples, collected from several ginseng planning localities [13]. Antcin K is the main triterpenoid of the endophytic fungus *Cinnamomum camphora*. The soil isolated the *Psychrobacillus* sp.AK 1817 strain dehydrogenated ergostene triterpenoid antcin K to produce antcamphin E and antcamphin F [37]. The pyrosporoid (diosgenin) were transformed into isonutigenone by *Streptomyces virginiae* isolated from the soil collected from the waste water pool of a steroid pharmaceutical factory [14]. The citral were transformed into acetone and hydroxy citronellal in 6 days by *Aspergillus niger*-PTCC 5011 isolated from the soil taken in Tehran prefecture [11]. The fungus *Lecanicillium attenuatum* W-1–9 was isolated from the deep-sea sediment of the polar region, which can transform xanthotoxin (*Radix Glehniae*) to lecaniside A-C [38]. Anthracene was transformed into 2,3-dihydroxynaphtha-lene, 3-hydroy-2 -naphthoic acid, and muconic acid by *Armillaria* sp. F022 isolated from decayed wood in a tropical rain forest [39,40]. Dehydroepiandrosterone was transformed into hydroxy-dehydroepiandrosterone by *Isaria farinosa* isolated from the bodies of dead insects in an ore cave [41]. *Salinivibrio costicola* GL6, which can convert caffeine to theobromine, was isolated from the Hoz Soltan Lake [42]. *Trichoderma harzianum* NJ01 could convert puerarin to 3-hydroxypuerarin, which was isolated from soil [43]. Consequently, the natural environment usually has a high content of natural compounds, and the natural selection of microorganisms enriches the strains with special transformation abilities.

### 2.2. Selection of Microbial Strains by Selective Medium

In order to isolate transformable strains from special environments or plants, the selection media of sole carbon or nitrogen sources is often used for further screening. For example, *Bacillus subtilis* zg-21, which can transform steroidal saponins, is isolated from the screening medium with the crude extract of total saponins from sisal as the only carbon source [36]. A white-rot fungus *Armillaria* sp. F022, which can convert anthracene, was isolated from the decayed wood in a tropical rain forest using an enrichment culture and with anthracene as the source of carbon and energy [39,40]. Using *Lantana camara* as the sole carbon source, a microorganism specific to LA was isolated from the soil. After nine subcultures and plate scribing, the bacterium *Alcaligenes faecalis* was isolated, which can transform lantadene a into its trans isomer lantadene X [44]. The soil was collected from the waste water pool of a steroid pharmaceutical factory, and the single carbon source and energy source was diosgenin. The *Streptomyces virginiae* was screened out from the EM medium supplemented with 1.5 g diosgenin and 10 mL soil-suspending liquid, which can transform the pyrosporoid (diosgenin) into the nutatigenin type sporosieroid (isonutigenone) [14]. The soil samples were collected from several ginseng plantation localities, and the *Paecolomies bainier* sp. 229 was screened out from the potato dextrose agar (PDA) plates containing 1 g/L Panax notoginseng leaves (PNL) [13]. *Micrococcus* sp. was isolated from rat feces by an enrichment technique using cholesterol as the sole source of carbon, which was able to convert cholesterol [45].

In addition, multiple metabolite detection methods are utilized to screen transformed microbial strains. For instance, 307 strains of bacteria that could grow on LB medium containing 0.4 mg/mL of puerarin were isolated from local soils, HPLC were used to analyze the transformed cultures, *Bacillus cereus* NT02 was screened out from 307 strains of bacteria, which can transform puerarin into puerarin-600-O-phosphate [46]. A total of 45 caffeine-degrading moderate halophiles were enriched from hypersaline lakes and the biotransformation of caffeine to theobromine was examined by TLC and HPLC. *Salinivibrio costicola* GL6 giving the highest yield of theobromine [42]. HPLC and TLC were used to monitor the transformation process during transfer. *Clonostachys rogersonia* was screened out from 16 plant pathogenic fungi, which can transform apophine alkaloids (*Stephania epigaea* LO) into 4R hydrophorphine alkaloids [47]. HPLC-DAD were used to analyze the transformed cultures. *Syncephalastrum racemosum* AS 3.264 was screened out from 15 different fungal strains, which can conduct the glucosylation of protostanes [28]. UPLC were used to determine the ability of the strain to digest antcin K, *Psychrobacillus* sp. AK 1817 was screened out from 4311 soil-isolated strains of bacteria, which can conduct the biotransformation of ergostane triterpenoid antcin K from *Antrodia cinnamomea* [37]. TLC were used to determine the ability to transform Huperzine A. *Ceriporia lacerate* hs-zjut-c13a was screened out from forty-nine fungal endophytes of *H. serrata*, which can conduct the biotransformation of hupA [28]. LC-ESI-MS/MS were used to determine the ability of the strain to transform non-steroid anti-inflammatory drugs (NSAIDs), 18 was screened out from 45 endophytic and epiphytic fungi (from Ascomycota), which can produce either mono- or dihydroxy derivatives of the NSAIDs [48]. HPLC were used for the screening of microbial cells for sterol side-chain cleavage. *Mycobacterium* sp. was screened out from 15 bacterial and fungal strains, which was able to convert *β*-sitosterol to 4-androstene-3,17-dione(AD) and 1,4-androstadiene-3,17-dione (ADD) [49]. All 38 Bifidobacterium strains were screened by *β*-glycosidase activity. Among them, five strains with high-level *β*-glucosidase activity, being active in the biotransformation of flavoid glycosides (daidzein, genestein, glycotein, and kaempferol) [50].

### 2.3. High-Throughput Screening of Transformed Microbial Strains

The traditional screening method has the problems of slow speed and large consumption, which cannot meet the needs of industrialization. The high-throughput screening method is a miniaturized, high-throughput screening technology. It can quickly and efficiently screen strains with high transformation ability from naturally selected or mutant selected strains [51]. Using biosensors specifically developed for screening specific microbial strains is an effective strategy and tool [52]. Using protein-based and nucleic-acid-based biosensors, through enzyme reactions, stain reaction and other methods to convert the test substance into absorbance and fluorescence signals to achieve the high-throughput screening of strains and their fermentation extracts. Its advantage is that the concentration of enzymes, nucleic acids, and metabolites in the fermented extract is converted into absorbance and fluorescence signals that are easy to detect through the reaction, which can be screened at a high speed and high throughput in microwell plates or even microfluidic droplets [53]. For example, Yang et al. used the determination wavelength and the reference wavelength of 6-hydroxynicotinic acid, which were were 251 nm and 231 nm, established a high-throughput screening method based on a 96-well microplate. Of the 1500 soil strains, four strains that efficiently use 6-hydroxynicotinic acid were screened [54]. Gao et al. used the determination wavelength of avermectins at 245 nm to quickly screen 38 strains that efficiently produced avermectins from 738 mutant strains in a 96-well microplate [55]. 

### 2.4. Selection of Microbial Strains by Bioinformation and Genomics

Bioinformatics and genomics are based on protein sequence, DNA sequence, and protein structure, which can provide scientific researchers with accurate and rapid genetic means to identify and screen microorganisms. In recent years, bioinformatics and genomics have shown great advantages in identifying new clusters. For example, Yu et al. used the engineering transcription factor BmoR linked to GFP fluorescent protein to screen butanol high-yield strains from multiple mutants by detecting the fluorescence intensity [56]. The higher concentration of malonyl-CoA in the strain is converted to flavonoids by biosensor RppA, resulting in dark red cultures. Yan et al. used colorimetric screening to analyze 1858 metabolic correlations from the *E. coli* library. Among the genes, 14 genes that significantly increased the yield of four malonyl-CoA-derived natural products (6-methylsalicylic acid, aloesone, etc.) were selected [57]. Qiu et al. constructed a sensor containing a functional gene encoding the erythritol degradation pathway and a green fluorescent protein gene and screened 1152 erythritol-producing mutant strains within one week by fluorescence detection. The best strain produced 148 g/L erythritol in a 3 L fermentor [58].

## 3. Optimization of Microbial Transformation Conditions

### 3.1. Solid Fermentation and Liquid Fermentation

Solid-state fermentation is a process. It is fermented in a medium without flowing water. Therefore, plant materials or compound powder are used as the medium substrate. Solid-state fermentation medium is rich in metabolites, enzymes, proteins, sugar, and sufficient water to maintain the growth and metabolism of microorganisms. Solid-state fermentation is more suitable for filamentous fungi, and the main factors affecting fermentation are humidity and temperature [9].

Several researchers studied cases of solid-state fermentation. For example, fresh *K. angustifolia* plant material was cut to a size of 1 cm and sterilized. An endophytic fungus *Umbelopsis dimorpha* SWUKD3.1410 grown on host-plant *K. angustifolia* and wheat bran and fermented in darkness at 28 °C without shaking for 50 days. After fermentation, many new compounds of terpenes have been produced [15] (Figure 1). Simultaneously, fermentation of *K. angustifolia* with endophytic fungus *Penicillium* sp. SWUKD4.1850 induced the production of rare highly oxygenated schitriterpenoids from nigranoic acid [21,22] (Figure 1). A total of 20 mg of aporphine alkaloid (*Stephania epigaea* LO) was added to the potato without excess water, which is used as a solid-fermentation medium. An endophytic fungus clonostachys rogersonia grown on the solid-state fermentation medium and fermented at 28 °C for 30 days. After sterilization, 4R-hydroyaporphine Alkaloids was isolated from fermentation products [47] (Figure 1). *Bacillus subtilis* can transform ginsenoside Rh1 and Re into gingsenoside Re after 10 days of solid-state fermentation (cubes of *Sanchi*) (Figure 1) [59]. The medium consisted of 250 g Oolong tea, 100 g wheat bran, 20 g glucose, and 5 g CaCO_3_ (Figure 1). After sterilization, the fungus *Paecilomyces gunnii,* grown on the solid-state fermentation medium and fermented at 25 °C for 35 days, ook caffeine (Oolong tea) and transformed it into theophylline and 1,7-dimethylxanthine [34].

In the process of liquid fermentation, plant extract or powder are added to ordinary liquid medium as a substrate, but the enzyme and protein activities of plants are damaged. Many reaction conditions such as substrate concentration, pH, temperature, and product concentration should be controlled in liquid fermentation [60].

For example, *Clonostachys rogersonia* was grown in a shaking culture at 28 °C in sterile PDB. The aporphine alkaloid (*Stephania epigaea* LO) dissolved in MeOH was added to the PDB, and incubation was continued for 7 days, resulting in very high conversions (99.5 ± 2.9%) of aporphine alkaloid converted to 4R-hydroxyaporphine alkaloids [47] (Figure 2a). The biotransformation of *β*-chamigrene-1*β*-ol (sesquiterpenoids in liverwort *R. hemisphaerica*) by *A. niger*. It was inoculated in Czapek peptone medium and cultivated in a rotary (100 rpm) at 30 °C for 3 days. Subsequently, the crude was separated to give three new metabolites, of which *β*-chamigren-1*β*,8*α*-diol was the major product (46.2% in isolated yield) [61] (Figure 2a). Coumarins dissolved in dimethyl sulfoxide was added to the Koch’·s K1 medium, *Cunninghamella elegans* ATCC 10028b and *Aspergillus brasiliensis* ATCC 16404 could convert coumarins 7-hydroxy-2,3-dihydrocyclopenta[c]chromen-4(1H)-one to 7,9-dihydroxy-2,3-dihydrocyclopenta[c]chromen- 4(1H)-one at 28 °C for 72 h with shaking at 120 rpm·min^−1^ [62] (Figure 2a). Glycyrrhetinic acid (*Glycyrrhiza glabra*) (5 mg) dissolved in Tween 80-H_2_O (0.5 mL; 1:3 *v*/*v*) added into the medium, and *Cunninghamella blakesleeana* could convert glycyrrhetinic acid to 3-oxo-7β-hydroxy-glycyrrhetinic acid (GA-1) and 7*β*-hydroxy-glycyrrhetinic acid (GA-2) at 30 °C for 5 days with shaking at 180 r·min^−1^ [63] (Figure 2a). An aqueous solution with powder of *Sanchi* (*Panax notoginseng*) was fermented by *Bacillus subtilis* and then cultured at 30 °C for 10 days on an orbital shaker at 120 rev/min. Ginsenoside Rh4 was produced in fermented *Sanchi*. Ginsenosides Rh1 and Re is the origin of ginsenoside Rh4 in fermented *Sanchi*, but adding singular compound ginsenoside Rh1 and Re into medium cannot produce ginsenoside Rh4 [59] (Figure 2b).

Liquid-state fermentation is generally more efficient and has a wide range of industrial applications. However, biotransformation does not always occur in both liquid-state fermentation and solid-state fermentation, and the products of solid fermentation and liquid fermentation are often different [17]. For instance, *Clonostachys rogersonia* can transform apophine alkaloids into 4R-hydrophorphine alkaloids after 30 days of solid-state fermentation (the potato containing 20 mg of aporphine alkaloid) (Figure 2c). Simultaneously, *Clonostachys rogersonia* can transform apophine alkaloids into 4R-hydrophorphine alkaloids after 7 days of liquid-state fermentation (the apophine alkaloids dissolved in MeOH and were added to the medium) [47] (Figure 2a). *Bacillus subtilis* can transform ginsenoside Rh1 and Re into Gingsenoside Re after 10 days of solid-state fermentation (cubes of *Sanchi*) (Figure 2d). Meanwhile, *Bacillus subtilis* can transform ginsenoside Rh1 into 3-O-*β*-D-glucopyranosyl-6-O-*β*-D-glucopyranos-yl-20(S)-protopanaxatriol after 4 days of liquid-state fermentation (the ginsenoside Rh1 dissolved in ethanol and was added to the medium) [53] (Figure 2a). *Chaetomium globosum* can transform panaxytriol into polyacetylene and cazaldehyde B after 7 days of solid-state fermentation (extract of *Red ginseng)* (Figure 2c), but metabolite A could not be isolated in this medium, which used PXT as a precursor [64] (Figure 2a).

Therefore, solid-state fermentation retains the enzyme activity of plant substrates, and, due to the space of solid substrate, compared with liquid fermentation, it provides uniform aerobic conditions for microorganisms, making metabolites more abundant, while liquid fermentation can make better use of substrate, and the culture time is short, which is more conducive to statistical and kinetic modeling [9,10].

### 3.2. Optimization of Fermentation Conditions

Once a good transformation strain is obtained, to obtain high yield of transformation products, including the substrate concentration, inoculation amount, time, temperature, pH, rotation speed, medium type, cell age, and other parameters must be optimized.

For instance, in the microbial transformation process of citral by *Aspergillus niger*-PTCC 5011, the major compounds produced in 6 days were acetone and hydroxy citronellal, while the major compounds after the 15-day period were acetone and citronellol [11]. *Trichoderma harzianum* can transform steroidal saponins (powder Of *Dioscorea zingiberensis* C.H.Wright) into diosgenin. By optimizing the fermentation conditions, the conversion efficiency is the highest at 30 °C, 4 days, pH 6, 33.33 g/L DZW. For example, the effect of temperature on the conversion efficiency was studied by setting three temperatures: the yield of diosgenin obtained at 30 °C was 85% and 27% higher than those obtained at 25 and 37 °C, respectively [12]. *Paecilomyces bainier* sp. 229 can transform Ginsenosides Rb1, Rb2, and Rb3 (protopanaxadiol type) into ginsenoside compound K by optimizing the fermentation conditions. The conversion efficiency is the highest at 30 °C, pH (4.5–5.5), 0.5% (substrate concentration), 5–7.5% (inoculum size), and 150–200 rev/min, with sucrose (carbon source) and soybean steep powder (nitrogen source). For example, the effect of the carbon source on the conversion efficiency is studied by setting four carbon sources (starch, lactose, sucrose, and maltose). Sucrose could increase ginsenoside compound K yield by about 10% as compared with a glucose-containing medium [13].

The cell age before adding the substrate is also a factor affecting microbial transformation. For instance, *S. virginiae* IBL-14 can transform diosgenin into diosgenone, when diosgenin was added to the 15 h old culture of *S. virginiae* IBL-14, 48 h after fermentation, the product reached the highest concentration, and the conversion efficiency reached 28.4%, while shorter or longer than 15 h, and the product concentration decreased significantly. By measuring the growth curve and pH, the biomass tended to remain stable after 12 h, the pH decreased to the lowest at 15 h, and then increased, indicating that cell age affects the transformation efficiency by affecting pH and biomass [14].

In addition to the basic optimization conditions such as temperature and pH, co-factors and surfactants can also promote microbial transformation by catalyzing the conversion process and promoting the solubility of the substrate. For instance, Escherichiacol (hyoscyamine-6*β*-hydroxylase) can transform hyoscyamine into 6*β*-hydroxycysteine and scopolamine. The enzyme activity of hyoscyamine 6β-hydroxylase depends on 2-oxoglutarate, ferrous ion and ascorbate are the co-factors of the reaction. When 2-oxoglutarate is absent, the conversion efficiency is only (4.53 ± 0.16)%. Adding ascorbate makes the conversion efficiency reach (90.35 ± 0.33)%, and adding Fe^2+^ makes the conversion efficiency reach (86.75 ± 2.41)% [7]. By optimizing the fermentation conditions, the addition of Fe^2+^ (below 1.25 mmol/L) and Tween-85 (0.07%) enhanced diosgenin yield by 22.07% and 5.50%, respectively [12]. Addition of MgSO_4_·7H_2_O and CaCl_2_ (0.1%, *w*/*v*) enhanced the conversion rate of ginsenoside compound K by 5% and 4%, respectively [13]. A total of 2% (*v*/*v*) n-tetradecane can accelerate the biotransformation of (R)-dicentrine to (4R,6*a*R)-4-hydroxydicentrine by clonostachys rogersonia because the conversion rate reaches 82.18% within 12 h when n-tetradecane is present. Without (R)-dicentrine, the same conversion rate will be achieved within six days [47].

The solubility and structural characteristics of substrates are also critical factors in microbial transformation. The water solubility of the steroids is low, and all kinds of solubilizers and surfactants are needed to ensure the transformation. Deep eutectic solvents (DESs), *β*-cyclodextrins derivatives (CDs), tween 80, Ionic liquids (ILs), and hydroxypropyl-*β*-cyclodextrin (HP-*β*-CD) are new classes of promising green solvents to overcome many drawbacks associated with organic solvents. In the process of the microbial transformation of steroids, they can promote the solubility of steroids, reduce product inhibition, enhance biocompatibility, and increase cell membrane permeability [65,66,67,68,69,70]. For example, *Mucor circinelloides lusitanicus* isolated from soil can introduce a 7a-hydroxyl group into substrates 1–5(5-en-3*β*-ol steroids), but substrates 6–7 (cholesteriol and diosgenin) cannot be transformed because C-17 has a special side chain. The large group at C-17 in substrates 6–7 made them unsuitable for the active site of the hydroxylase [35]. *Mycobacterium sp*. NRRL B-3683 and *Mucor sp*.NRRL B-3805 were able to convert *β*-sitosterol, cholesterl, stigmasterol, and ergosterol to 4-androstene-3,17-dione (AD) and 1,4-androstadiene-3,17-dione (ADD). Among them, the conversion rate of *β*-sitosterol is the highest, reaching 75.87% and 81.83%, which is much higher than that of other substrates. At the same time, the solubility of *β*-sitosterol is almost the same as that of other substrates, indicating that the conversion rate of sterol to androsterones is more related to the structural characteristics of the substrates, which depends on the ability of enzyme and substrate interaction [49]. 

More examples of optimal fermentation conditions for microbial transformation are presented in Table 1 (Appendix A).

### 3.3. Fermentational Strategy

A number of fermentational strategies can be found in the article describing the improvement of the efficiency of microbial transformation. Fed-batch fermentation helps prevent the influence of high concentration substrate restriction and inhibition by controlling the substrate concentration below the toxic level [74,75]. The cassava flour was fermented to product glutamic acid by *Brevibacterium divarium*, but the high initial concentration (>300 g/L) of cassava flour inhibited cell growth and glutamic acid production. The maximum production of glutamic acid by *Brevibacterium divarium* was only about 4 g/L, compared with 116.2 g/L in fed-batch fermentation [76]. Coumarin (tonka bean meal) was transformed into melilotic acid by *Saccharomyces cerevisiae*, but coumarin > 0.5 g/L was toxic to yeast cells, thus the yield of melilotic acid was reduced. However, the product melilotic acid was non-toxic. When coumarin was reduced to a lower level at 155 h, the addition of coumarin increased the concentration of melilotic acid from 0.5 g/L at 155 h to 0.90 g/L at 312 h [72]. Phytosterol is transformed into 1,4-androstadiene-3,17-dione (ADD) by *Mycobacterium neorium*. The strategy of semi-batch fermentation moves the enzyme of each fermentation stage into a high activity state. In the start-up stage, the yield of boldenone tended to be stable after four days. In the semi-batch fermentation stage, 50% broth was withdrawn every two days, and then the same volume of medium containing phytosterol was added. After three cycles of semi-batch fermentation, the yield rate of ADD was 9.29 times higher than that in the start-up stage (0.17 g/L/d), Additionally, the time needed to obtain the same yield was shortened by 7.5 d compared with the common fermentation. At the same time, glucose supplement can increase the output of ADD converted from boldenone by *Pichia pastoris*. The ratio of [NADPH]/[NADP^+^] in cells can be stabilized by adding glucose of appropriate concentration. When the glucose water supply is 50 g/L, the yield of BD is still stable after reaching 76% and increased by 11% [77]. *Saccharomyces cerevisiae* used pyruvate to synthesize lycopene, but excessive ethanol accumulation in the cell growth period will lead to a decrease in lycopene accumulation. Using a two-stage fed-batch fermentation strategy, the fermentation can be subdivided into a cell growth period and a lycopene production period, respectively, to achieve high cell density and high lycopene production. In the first glucose feeding stage, glucose was controlled below 1 g/L, and residual ethanol was controlled below 10 g/L. In the second feeding stage, the feed solution containing ethanol as carbon source was added to stimulate the production of lycopene, and the concentration of ethanol was controlled below 5 g/L. Finally, 3.28 g/L of lycopene was obtained [78].

## 4. Microbial Transformation Based on Synthetic Biology

### 4.1. Heterologous Biosynthesis Based on Gene Engineering and Enzyme Engineering

The milestone of microbial transformation industrialization is that the *American puqiang* pharmaceutical factory transformed progesterone into 11*α*-hydroxyprogesterone by using the hydroxylase of *Aspergillus niger* in the 1950s. Then, with the development of gene recombination technology, several different synthetic genes were constructed into engineering strains to realize multi-step transformation reactions to enrich the function and application of microbial transformation (Figure 3).

The earlier reported microbial transformation with the use of heterologous expression strains was in the 1990s. Since then, more authors use heterologous expression strains to obtain high-yield compounds. Geerlings et al., in 1998, cloned cDNAs’ coding for strictosidine synthase (STR) and strictosidine *β*-glucosidase (SGD) from the medicinal plant *Catharanthus roseus*, and then connected to vector pYPGE15 and transferred it into *Saccharomyces cerevisiae*; secologanin and tryptamine were completely hydrolyzed to cathenamine by *S. cerevisiae* [79]. After that, Shen et al. expressed strictosidine synthase (*Rauwolfia serpentina*) in 22 strains of bacteria (*Aeromonas* sp., *Bacillus licheniformis*, *Klebsiella oxytoca*, etc.) in 2001, all of which produced strietosidine aglyeone by 3α(s)-strictosidine deglycosylation and rearranged the alkaline to vallesiachotamine and isovallesiachotamine [80]. Seeger et al. in 2003 constructed plasmids containing the biphenyl-2,3-dioxygenase (BphA) and biphenyl-2,3-dihydrodiol 2,3-dehydrogenase (BphB) genes, then it was transferred to *Burkholderia* sp. strain LB400. BphA can convert 7-hydroxy-8-methylisoflavone and 7-hydroxyysoflavone into DHD, and BphB can convert DHD into 7,2′3′-trihydroxy-8-methylisoflavone and 7,3′4′—trihydroxyysoflavone [81]. In addition, Kanako et al. also utilized the yeast-expressing plant-membrane-bound prenyltransferase SfN8DT-1 to convert naringenin to its prenylated form (8-dimethylallylnaringenin) in 2009 [6]. More recently, Zhou et al., firstly, uses the high-yield β-carotene producing *Saccharomyces cerevisiae* strain for rapid cell growth and the accumulation of precursor *β*-carotene in 2015, and then the *β*-carotene ketolase and hydroxylase gene had been constructed in the strain to induce the production of the target compound astaxanthin, which increased the yield by 13-fold higher than the first astaxanthin-producing *S. cerevisiae* strain [82]. In 2017, Cardillo et al., using cDNAs of hyoscyamine-6*β*-hydroxylase (H6H), amplified from *Brugmansia candida*, and then connected it to pET32a vector(+) and transferred it to *E. coli*. Both protein extracts and whole cells of the induced bacteria can hydroxylate the hydroxyamine to produce 6*β*-hydroxyamine, and then dehydrogenate the 6*β*-hydroxyamine to generate scopolamine [7]. In the past few decades, a series of heterologous expression methods were used to achieve microbial transformation. However, the conversion efficiency was the problem still exists, such as low yield of the desired product, a cytotoxic by-product of the metabolism, a complex synthesis process. More examples of microbial synthesis and transformation by heterologous expression are presented in Table 2 (Appendix A).

### 4.2. Reduce Cytotoxicity through Metabolic Engineering

The products produced by exogenous expression genes are often cytotoxic to the host, which can inhibit the growth of cells and thus reduce the yield.

One solution is to mutate synthetic genes using metabolic engineering to regulate the metabolic flow of intracellular synthetic precursors. Different synthesis pathways are located in distinct organelles, which can prevent toxic products and intermediates from entering the cytoplasm. Farnesyl diphosphate (FDP), located in the cytoplasm, is the precursor of sterols, ubiquinones, and terpenes. By blocking the FDP gene synthesizing other substances in Saccharomyces cerevisiae through mutation, FDP flows into mitochondria in a large amount, thus increasing the yield of sesquiterpene value and amorphadiene by an 8- and 20-fold rate, respectively [107]. 

In other studies, reducing the toxicity of metabolites in the early growth process by the use of the dynamic regulation of the growth of strains was documented. Vanillin and phloroglucinol, the products of microbial transformation, have toxic side effects on host bacteria and can destroy the integrity of biofilm, especially in the early stage of microbial growth. Liang et al. used *E. coli* as a host and ferulic acid as the substrate to catalyze the formation of vanillin via tansferuloyl-CoA synthetase (Fcs) and enoyl-CoA synthetase (Ech). Firstly, Liang et al. designed and constructed a dynamic regulatory element based on a hucr mutant by first screening the specificity of regulatory protein recognition effectors. Then, they regulated the expression of Fcs and Ech genes to reduce metabolic burdens in early growth and improve the metabolic activity in late growth. When the concentration of vanillin was high in the later stage, the tolerance of high-density cells to vanillin increased. In addition, Laing et al. also designed the use of vanillin adsorption resin to adsorb vanillin in fermentation broth or improve the tolerance of strains to vanillin [8].

### 4.3. Increase Metabolic Flux of Products through Metabolic Engineering

With the continuous development of molecular biology, genetic engineering, metabolic engineering, and other disciplines, gene engineering and metabolic processes are often combined to analyze, turn on, or turn off gene expression at an appropriate time to realize the dynamic regulation of modern metabolic pathways, as to improve the specific transformation of engineering strains and the production capacity of metabolites. Common methods include increasing the supply of precursors, weakening the flux of competitive pathways, and reducing the inhibition of products. In several latest studies, several authors have contributed to expanding the strategy of increasing metabolic flow through metabolic engineering.

Pei et al. introduced a c-glucosyltransferase (Gt6CGT) gene from *G. triflora* to an *E. coli* strain for converting luteolin to isoorientin. However, acetic acid is the by-product of *E. coli*. It can reduce the expression level of recombinase and inhibit the growth of bacteria. A total of 1% Glycerol as the carbon source could decrease acetic acid accumulation; then, introducing the cellobiose phosphorylase gene (cep) and the UTP-glucose-1-phosphate uridylyltransferase gene (ugpA) to *E. coli* results in more flow of cellobiose to UDP glucose and less flow to TCA glucose. Through increasing UDP glucose supply and controlling acetic acid to significantly increase the output of isourientin, the highest yield of isourientin was 1371 mg/L [108].

Wang et al., firstly, found that the polyketide acti norhodin (Act) was mainly produced in the later stage of *Streptomyces*, but the yield was relatively low. Additionally, they found that intracellular triacylglycerols (TAGs) can regulate more carbon flow to polyketide synthesis by providing reducing power for polyketide synthesis and changing the level of reducing power after the bacteria enter a stable period. Then, through genome data mining and physiological and biochemical analysis, a series of genes related to fatty acid anabolic metabolism were identified. Additionally, using comparative analysis to identify the metabolic model of genetic influence. Finally, by controlling acyl-CoA synthetase Sco6196, the yield of the Act was 190% higher than that of control. The results showed that the regulation of metabolic flow was important to improve the production of secondary metabolites [18].

Tomas et al. used STC5 as the screening probe and found a sesquiterpene synthase FgJ02895 from *Fusarium grainearum* J1-012, which can efficiently synthesize guaia-6,10(14)—diene and expressed it in *E. coli* and *Saccharomyces cerevisiae*. At the same time, through the metabolic process to strengthen the mevalonate pathway and the downstream synthesis pathway of guaia-6,10(14)-diene and then through the construction of a series of mutants, they enhanced the supply of the upstream substrate and the downstream synthesis pathway of guaia-6,10(14)-diene to achieve a high yield of guaia-6,10(14)-diene [19].

Purine is synthesized with 5-phosphoribosylpyrophosphate (PRPP) and glutamine as precursors, but the synthesis of purine compounds is closely controlled by many levels, which are difficult to accumulate in the natural state. Liu et al. analyzed the synthesis regulation mechanism of purine metabolism pathway using transcriptomics and fluorescence quantitative PCR. The accumulation of hypoxanthine was increased nearly ten times using the deregulation of PurR (regulatory proteins) and site-directed mutation of the key enzyme to alleviate substrate feedback inhibition, increase the accumulation of purine synthesis precursor, and destroy the branch metabolism pathway of IMP to AMP and GMP [109]. More examples of optimization and control strategies through metabolic engineering are presented in Table 3 (Appendix A).

### 4.4. Multi-Strain Collaborative Biotransformation Strategy

In nature, due to the existence of microorganisms in the form of community, and the existence of different microorganisms with complementary metabolism, there is generally collaborative microbial transformations [113]. A single strain has a series of problems such as excessive strain burden on strains, limited advantages of single platform strains, and long construction times of complex functional strains. Compared with a single strain, the synthetic microbiome uses a single strain as each module of the assembly line and improves the synthesis efficiency of the target product through the division and cooperation of multiple strains [114]. The method of synthetic biology is used to design modular metabolic pathways, then the corresponding strains are constructed for each metabolic module, and, finally, a multi-strain co-culture system is established to synthesize the target product.

For instance, we established a step wise biotransformation of boldenone (BD) from phytosterol (PS) by combining *Mycobacterium neoeurum* and *Pichia pastoris*. Firstly, C1,2 dehydrogenation of phytosterol was realized by *Mycobacterium neorium*. Finally, phytosterol was transformed into 1,4-androstadiene-3,17-dione (ADD), and then ADD was transformed into boldenone by 17 *β*-carbon reduction of *Pichia pastoris* [77].

In a similar study, (R)-(-)-mandelic acid was a key intermediate for the synthesis of penicillin, *Saccharomyces cerevisiae,* and *Bacillus cereus* was a catalyst at the first step and the second step, respectively, and (R)-(-)-mandelic acid was synthesized by two-step biotransformation. Firstly, (R)-(-)-mandelic acid ethel ester was synthesized by asymmetric reduction of ethel benzoylformate by *Saccharomyces cerevisiae*, and then it was converted to(R)-(-)-mandelic acid through hydrosis by *Bacillus cereus* [115].

In another study, a high concentration of gamma aminobutyric acid (GABA) was produced by cassava and other crops. Firstly, the glutamic acid was produced by the fermentation of the liquid cassava powder with *Corynebacterium glutamicum*, followed by the transformation of glutamic acid to GABA with resting cells of *L. plantarum* GB01-21, and the maximum productivity of GABA was 2.68 g/L/h [76].

Heterologous expression strains of the same microorganism can also form a synthetic microbiome. Li et al. used three metabolically engineered *E. coli* strains for co-cultivation of heterogeneous biosynthesis of complex natural product rosmarinic acid (RA). The three strains were used as CA (synthesis of caffeic acid from 4-hydroxyphenylpyruvate) module, an SAA (synthesis of salvianic acid from 4-hydroxyphenylp-yruvate) module, and an RA (synthesis of rosmarinic acid from caffeic acid and salvianic acid) module. Finally, 172 mg/L of RA was produced, which was 38 times higher than the single-cultured parent strain [116].

## 5. Combination of Chemical Semisynthesis and Microbial Transformation

Many natural medicinal products can be directly separated from plants or synthesized organically. However, some natural products of terpenes, due to their complex structure, limited sources of raw materials, low content, complex synthesis process, environmental pollution, and other reasons, limit the development and utilization of medicinal natural product resources. The combination of microbial synthesis and organic synthesis provides an opportunity for natural products that are difficult to synthesize directly [19].

Microorganisms can transform chemically synthesized products into new structures and active products. For example, the reaction of 7-hydroxycoumarin with epichlorhydrin in the presence of K_2_CO_3_ led to the formation of oxirane derivative. The coumarin 7-(3-(cyclopropylamino)-2-hydroxypropoxy)-4-methyl-2H-chromen-2-one was synthesized via the nucleophilic opening of the oxirane ring by cyclopropyl amine. Then, the coumarin was transformed into 7-(3-Cyclopropylamino-2-hydroxy-propoxy)-4-meth-oxymethyl-chromen-2-one by *Candida albicans*. Ultimately, inactive chemical semisynthetic substrates are converted into products with strong antibacterial activity and cancer cell toxicity [117]. In another recent study, 6*β*-Acetoxy-1*β*,4*β*-dihydroxyeudesmane was isolated from *Sideritis leucantha* Cav. subsp. meridionalis, the first step was catalyzed by KOH/MeOH and CrO_3_/H_2_SO_4_, the second step is catalyzed by SOC_l2_/Py, and, finally, 1-oxoeudesman-4*β*,6*β*-diyl-S(R)-cyclic sulfite and 1-oxo-eudesman-4*β*,6*β*-diyl-S(S)-cyclic sulfite are generated. 1-oxo-eudesman-4*β*, 6*β*-diyl-S(S)-cyclic sulfite was hydroxylated and acetylated by Rhizopus nigricans to 8*α*,11-dihydroxy-1-oxoeud-esman-4*β*,6*β*-diyl-S(S)-cyclic sulfite and 4*β*,6β,8*α*,11-tetrahydrox-yeudesman-1-one. The metabolic mixture transformed by microorganisms was then catalyzed by Ac2O/Py to produce 8*α*-acetoxy-1-oxoeudesman-4*β*,6*β*-diyl-S(S)-cyclic sulfite and 11-hydroxy-1-oxoeudesman-4*β*,6*β*-diyl-S(S)-cyclic sulfite [118].

Furthermore, the combination of microbial synthesis and organic synthesis can be used for targeted synthesis of products, replacing the complicated chemical synthesis of steps. Englerin A is a plant sesquiterpenoid compound with significant anti-renal cancer activity. Tomas et al. took guaia-6,10 (14)-diene as the substrate. First, intermediate **3** was obtained under the catalysis of PhSiH3, then intermediate **4** was obtained by selective dihydroxylation with Sharpless asymmetric catalyst, then **4** is oxidized by dimethylperoxyketone to obtain product **7**. Intermediate **8** was obtained by adding glacial acetic acid to **7**, which is then esterified with cinnamic acid. Finally, the target product englerin A (yield 85%) was obtained by TBAF reagent [19]. Kainic acid is a neurotoxin isolated from seaweed, its chemical synthesis requires at least six steps with a yield of less than 40%. Chekan et al. identified and cloned dskabA (n-isopentenyltransferase) and dskabC genes (*α* kg dependent dioxygenase) from *Digenea simplex*, then expressed and purified them in *E. coli*. DskabA synthesized prekainic acid with dimethylallyl pyrophosphate and L-glutsmic acid as substrates, then dskabC transformed prekainic acid into kainic acid. After the improvement of the process, L-glutsmic acid and formic acid react under the catalysis of 50% MeOH and NaBH_4_ to form prekainic acid with a yield of 56%. Then, *E. coli* overexpressing dskabC gene transformed the prekainic acid into kainic acid, the yield of 57% was higher than that of dskabC (46%) [119].

## 6. Detection of Transformed Metabolites by Multi-Omics

Conventional methods for detecting metabolites include HPLC, GC-MS, NMR, etc. Furthermore, an increasing number of studies transferred to the analysis of specific secondary metabolites in the microbial community, as well as the study of the biological effects of metabolite changes. Combined with genome data mining, metabonomics analysis, and physiological and biochemical indicators, not only can we obtain the best time and highest yield of the target product, but also can elucidate the biosynthesis pathway and function of the target product.

In 2016, Wu et al. collected the fermentation broth of *Streptomyces* sp. mbt76 every 24 h for 5 days. The extract was analyzed by ^1^H-NMR spectroscopy, and the data were analyzed by principal component analysis (PCA) and the latest structures (OPLS). The results showed that 4 days was the best time to accumulate antibacterial substances, among which acetyltryptamine and trimethylated isocoumarin were the main antibacterial substances, and methoxylation improved the antibacterial activity of coumarin [71]. In 2017, Martins et al. used *Penicillium janczewski* hydroxylated labdanolic acid (plant terpenoids) to generate 3*β*-hydroxy-labdanolic acid, and analyses of mycelial and extracellular differential proteomes demonstrated that the plant terpenoid increased stress responses, especially against oxidative stress (e.g., accumulation of superoxide dismutase) and, apparently, altered mitochondria functioning. The combination of proteomics can be utilized to study how plant natural products affect microbial metabolism in the process of microbial transformation [120]. More recently, Zhao et al. analyzed the related enzymes and their networks involved in polysaccharide degradation and phenol metabolism by HPLC, metabonomics, and proteomics in each stage of microbial fermentation of Pu’er tea. It was found that 19 enzymes, such as glycoside hydrolase, phenol 2-monooxygenase, and salicylaldehyde dehydro genase were related to the oxidation, transformation or degradation of phenolic compounds [121]. Compared with single microorganisms, the analysis of metabolites in a microbial community is difficult. Cao et al. constructed the correlation network between microorganisms and metabolites by detecting the characteristic small molecules of specific microorganisms in microbial community and locating them on the system development. By combining the tandem mass spectrometry data with the sub genomic data, the microorganism responsible for the production of corynomycolenic was precisely discovered. By mining the genomic data, the strain DSM20755 with the gene of corynomycolenic was found, which can synthesize the biosynthetic enzyme of corynomycolenic [122]. With the development of multi-omics, the change in chemical composition and metabolic pathway was detected in the process of fermentation and transformation, as to realize the control and optimization of the fermentation process.

## 7. Concluding Remarks and Future Perspectives

Microbial transformation has been studied for centuries. It has wide applications and great market potential in mining new compounds, but its application and development are limited by its low yields. The selection of strains and the optimization of fermentation conditions have many advantages, resulting in high-yield conversion products. As the continuous development of novel strategies and tools in metabolic engineering and multi-omics, the use of systems biology approaches can achieve targeted regulation of metabolic pathways and flow through over-expression, down-regulation, and deletion, leading to reduced product inhibition and increased goals product yield. Among other various strategies, the combination of chemical semi-synthesis and microbial synthesis can provide opportunities for the synthesis of compounds with complex structures. The multi-step catalytic reaction of multiple strains can also guide the synthesis of complex products. In addition, the multi-omics detection method can make the conversion process of the conversion products more accurate. Finally, many outstanding questions still exist. These include: (i) How do we obtain microbial strains through high-throughput screening? (ii) What is the relationship and choice between chemical synthesis and microbial synthesis? (iii) Can we find the connection between the transformed microbial strains in the synthetic microbiome? In general, these methods can transform microorganisms towards the directions of targets, high efficiency, and environmental sustainability.

## Figures and Tables

**Figure 1 molecules-28-02619-f001:**
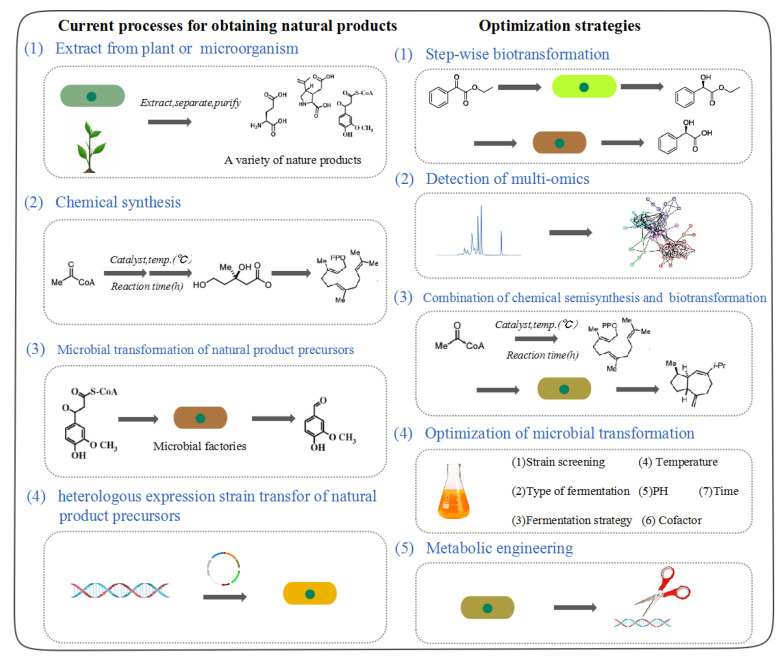
Current processes for obtaining natural products: (1) Extract natural products from plants or microorganisms. (2) Chemical synthesis of natural products. (3) Microbial transformation of natural product precursors. (4) Transformation of natural product precursors by heterologous expression strains. Multiple optimization strategies: (1) Multi-step microbial transformation of natural products. (2) Multi-omics detection of natural products transformed by microorganisms. (3) Combination of chemical semisynthesis and biotransformation. (4) Optimize the conditions of microbial transformation process, including strain screening, type of fermentation, fermentation strategy, temperature, pH, and co-factor. (5) Metabolic engineering changes the transformation ability of microbial strains for the purpose of alleviating feedback inhibition, reducing cytotoxicity, increasing metabolic flow, reducing by-products, and increasing precursor availability.

**Figure 2 molecules-28-02619-f002:**
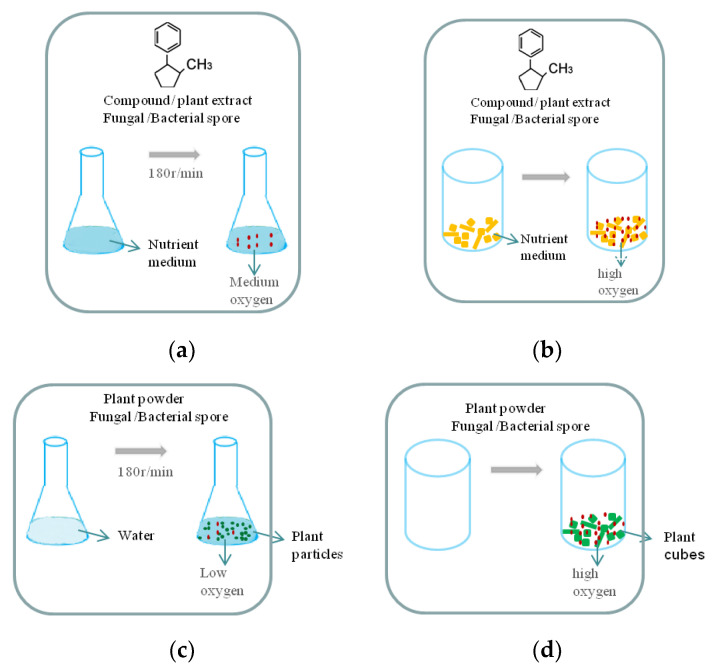
Schematic diagram of fermentation method: (**a**) Liquid fermentation: Nutrient medium (LB, TSB, PDB, etc.) added with compound/plant extract and fungal/bacterial spore at 180 r·min^−1^. (**b**) Liquid fermentation: Water added with plant powder and fungal/bacterial spore at 180 r·min^−1^. (**c**) Solid state fermentation: Nutrient medium (wheat bran, potato cubes, etc.) added with compound/plant extract and fungal/bacterial spore. (**d**) Solid state fermentation: Plant cubes added with compound/plant extract and fungal/bacterial spore.

**Figure 3 molecules-28-02619-f003:**
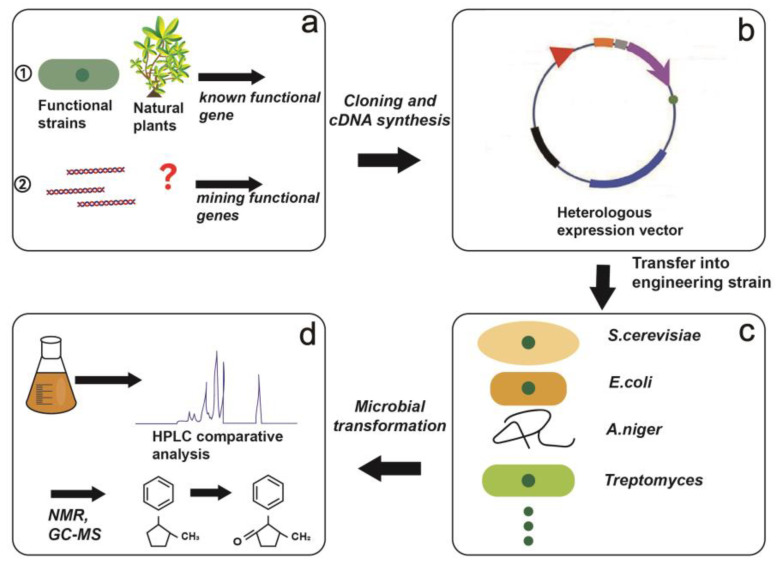
Basic steps of heterologous biosynthesis. (**a**) ① Select known functional genes from plants and strains. ② Screen the unknown functional genes of the strain through the analysis of genome and transcriptome, and then select the homologous genes of other strains in NCBI, and then select the highly expressed functional genes through the level of transcription and transformation products. (**b**) Clone and synthesize cDNA, then select the appropriate vector and construct. (**c**) Transform the recombinant plasmid into the engineering strain. (**d**) Use the engineering strain to ferment the target substrate, HPLC comparative analysis to detect the transformation ability, and then analyze the structure of the transformation product by NMR and GC-MS.

**Table 1 molecules-28-02619-t001:** Examples of fermentation conditions for microbial transformation.

Substrate	Solvent	Microbial Catalyst	Cell Age	Conditions	Obtained Compounds	Conversion Yields	Ref.
Citral	Methanol	*Aspergillus niger*-PTCC 5011	2 day	27 °C; pH 5.5; 6 days, 15 days; 150 r·min^−1^	6 day:hydroxy citronellal 15 day:citronellol	37.0%; 36.0%	[11]
Genistein	DMSO	*Streptomyces* sp. MBT76	2 day	30 °C; pH 6.8; 3 days; 220 r·min^−1^	Methoxylated isoflavones	-	[71]
Coumarin (tonka bean meal)	EtOH	*Saccharomyces cerevisiae*	1 day	30 °C; pH 4.5–6.5; 150 h; 180 r·min^−1^	Melilotic acid	3.6%	[72]
Hyoscyamine	-	*Escherichiacol* (Hyoscyamine-6β-hydroxylase)	-	30 °C; pH 7.8; 25 h; 250 r·min^−1^; FeSO4, Ascorbate,2-oxoglutarate	6-hydroxyhyoscyamine; Scopolamine	5.22%; 0.67%	[7]
Steriodal saponin	Water	*Trichoderma harzianum*	1 day	30 °C; pH 6; 4 days; 150 r·min^−1^; Na_2_HPO_4_, KH_2_PO_4_, 0.93 mmol/L Fe^2+^; 0.07% (*w*/*v*) Tween-85	Diosgenin	30.05%	[12]
GinsenosidesRb1, Rb2, Rb3	Water	*Paecilomyces bainier* sp. 229	3 day	28 °C; pH 4.5–5.5; 3 days;150 r·min^−1^; Mg,Ca; 0.2% Tween80	Ginsenoside compound K	82.6%	[13]
Sitosterol; cholesterol; stigmasterol; ergosterol	Tween 80	*Mycobacterium* sp.	2 day	25 °C; 20 days; 200 r·min^−1^	4-androstene-3,17-dion; 1,4-androstadiene-3,17-dione; testosterone	81.83% (total)	[49]
Dihydroalisol A; alisol G, F, B	Acetone	*Syncephalastrum racemosum* AS 3.264	1 day	4 days	11-OH-Glucosylation(1a; 2a; 3a; 4a)	25%; 58%; 55%; 53%	[28]
Labdanolic acid	Ethanol	*Penicillium janczewskii*	-	60 days; 0 r·min^−1^	3β-hydroxy-labdanolic acid	>90%	[73]
Thymoquinone	Acetone	*Aspergillus niger*	3 day	28 °C; 7 days; 128 r·min^−1^	5-isopropyl-2-methyloxepin-1-on; 3-hydroxy-5- isopropyl-2-Methylcyclohexa-2,5-diene-1,4-dione; 5-isopropyl-2-methylbenzene-1,4-diol	4.9%;7.3%; 28.0%	[73]
Diosgenin	Heated ethanol	*S.virginiae* IBL-14	15 h	30 °C; pH 5.2; 2 days; 200 r·min^−1^	Diosgenone; Isonuatigenone	28.4%	[14]
Ginsenoside Rh1,Re	Water	*Bacillus subtilis*	3 day	30 °C; pH 7; 10 days; 120 r·min^−1^	Gingsenoside Rh4	-	[53]
Naringenin	-	Yeast (SfN8DT-1)	-	30 °C; 20 h	8-dimethylallylnaringenin (8DN)	1.9%	[6]
(R)-dicentrine	MeOH	*Clonostachys rogersonia*	3 day	28 °C; 7 days; 2% (*v*/*v*); n-tetradecane	Corresponding (4R,6aR)-4-hydroxydicentrine	99.5%; 97.4%; 99.9%	[43]
Nigranoic acid	-	*Umbelopsis dimorpha*, *Penicillium* sp.	3–5 day	28 °C; 25 days, 45 days; Solid-state	A series of high oxidation triterpenes	-	[15,21,22]
Caffeine	-	*Paecilomyces gunnii*	-	25 °C; 35 days; Solid-state	Theophylline; 1,7-dimethylxanthine	-	[34]

**Table 2 molecules-28-02619-t002:** Products precursor heterologously expressed genes and sources engineering strains.

Products	Precursor	Heterologously Expressed Genes and Sources	Engineering Strains	Ref.
Neu5Ac	N-acetylglucosamine (GlcNAc), pyruvic acid	GlcNAc epimerase and Neu5Ac al-dolase	*E. coli*	[83]
CMP-Neu5Ac	Neu5Ac	CMP-Neu5Ac synthetase (*N.meningitidis*)	*E. coli*	[84]
Vallesiachotamine, Isovallesiachotamine	Tryptamine, Secologanin	Strictosidine synthase (*R.serpentina*)	*Aeromonas sp., Bacillus licheniformis,*	[80]
Cathenamine; Strictosidine	Tryptamine, Secologanin	STR, SGD (*C. roseus*)	*Saccharomyces cerevisiae*	[79]
7,2′3′-trihydroxy-8-methylisoflavone, 7,3′4′-trihydroxyysoflavone	7-hydroxy-8-methylisoflavone, 7-hydroxyysoflavone	BphA, BphB	*Burkholderia sp.*LB400	[81]
Resveratrol, Piceatannol, Isorhapontigenin	4-coumaric acid, Caffeic acid, Ferulic acid	4CL1 (*A. thaliana*),STS (*A. hypogaea*)	*E. coli*	[85]
Naringenin, Pinocembrin, Resveratrol	4-coumaric acid, cinnamic acid	CCL (*S. coelicolor*), CHS (*A. thaliana*)	*S. venezuelae*	[86]
Amorphadiene	Valencene	SfN8DT-1	*Yeast*	[6]
Linalool	Mevalonic acid	LIS (*C. brewer*), LIS (*L. angustifolia*)	*Saccharomycesce. cerevisiae*	[87]
(S)-reticuline	(S)-norcoclaurine	TYR (*S.castaneoglobisporus*), DODC (*P.putida*), 6OMT, CNMT, 4′OMT (*C.japonica*), MAO (*M. luteus*)	*E. coli*	[88]
Astaxanthin	Isopentenyl diphosphate	CrtW148 (*N. punctiforme*), crtE, crtB, crtI, crtY, crtZ (*P.ananatis*)	*E. coli* BW-ASTA	[89]
Resveratrol	*p*-coumaric acid, Malonyl-CoA	4CL1 (*A. thaliana*), STS (*V. vinifera*)	*E. coli*	[90]
7-*O*-Methylaromadendrin	*p*-coumaric acid	4CL (*P. crispum*), CHS (*P.hybrida*), CHI (*M. sativa*)	*E. coli*	[91]
Miltiradiene	Glucose	CPS, KSL, BTS1, ERG20 (*S. miltiorrhiza*)	*S. cerevisiae*	[92]
Perillyl alcohol	Limonene	GPPS (Abies grandis), LS (Mentha spicata), P450 (co) (*Mycobacterium* HXN 1500)	*E. coli*	[93]
Artemisinic acid	Amorphadiene	CYP71AV1, CPR1, CYB5, ADH1, ALDH1 (*A. annua*)	*S. cerevisiae*	[94]
Ferruginol	Miltiradiene	CYP76AH1 (*A.thaliana*)	*S. cerevisiae*	[95]
*β*-carotene	Pyruvate, glyceraldehyde-3-phosphate	CrtEXYIB (*P. agglomerans*)	*E. coli*	[96]
Catechin, Afzelechin	Eriodictyol, Naringenin	F3H (*C. sinensis*), DFR (*A.andraeanum*), LAR (*D. uncinatum*)	*E. coli*	[97]
Pinostilbene, Resveratrol	Tyrosine	TAL (*S. espanaensis*), 4CL(*S. coelicolor*), STS (*A. hypogaea*), SbOTM1, OTM3 *(S. bicolor*)	*E. coli*	[98]
Morphine, 14-hydrocodine,	Thebaine	T6ODM, CODM, COR (*P.somniferum*)	*S. cerevisiae*	[99]
Astaxanthin	Carotene	CrtZ and BKT (*H. pluvialis*)	*S. cerevisiae*	[82]
Berberine	Norlaudanosoline	6OMT, 4′OMT, BBE (*P.somniferum*), S9OMT (*T. flavum*), CAS (*T.flavum*), CPR (*A. thaliana*)	*S. cerevisiae*	[100]
Sanguinarine, Stylopine	Norlaudanosoline	ATR1 (*A. thaliana*), CFS, STS, P6H (*E. californica*), 6OMT, OMT, TNMT (*P. somniferum*)	*S. cerevisiae*	[101]
Ginsenosides Rh2,Rg3	Protopanaxadiol	PPD, UGT (Panax ginseng)	*S. cerevisiae*	[102]
Thebaine	(R)-reticuline	ATR2 (*Arabidopsis thaliana*), CPR (*Papaver somniferum*), CPR (*Rattus norvegicus*)	*E. coli*	[103]
Nocapine, Noscapine	Norlaudanosoline	CYP82Y1, TNMT, MT1, MT2, MT3, SDR1 (*P. somniferum*), CAS (*C. japonica*), ATR1 (A. thaliana)	*S. cerevisiae*	[104]
Dammarenediol-II	Farnesyl diphosphate	SS, SE, CPR (*S. cerevisiae*), SE (*M.capsulatus*) CPR (*A.thaliana*)	*E. coli*	[105]
Scopolamine	Hyoscyamine	H-6-H	*E. coli*	[7]
(S)-reticuline 3-O-sulphate (S)-reticuline 7-O-sulphate	(S)-reticuline	HSULT1A1, hSULT1A3, hSULT1E1	*E. coli*	[106]

**Table 3 molecules-28-02619-t003:** Examples of optimization and control strategies through metabolic engineering.

Products	Precursor	Heterologously Expressed Genes and Sources	Engineering Strains	Ref.
Valencene and amorphadiene	FDP	Deregulate HMG1, express mitochondrion-targeted FDPS, block other metabolic pathways of FDP, FDP flows into mitochondria in a large amount	*S. cerevisiae*	[107]
Vanillin	Ferulic acid	Dynamic regulatory element (Express Fcs, Ech, hucr mutant), reduces the metabolic burden and toxic side effects during early growth	*E. coli*	[8]
Isoorientin	Luteolin	Expression of synthetic gene Gt6CGT. Express Cep, ugpA to inhibit accumulation of by-product acetic acid, more flow of cellobiose to UDP glucose and less flow to TCA	*E. coli*	[108]
Acti norhodin	TAGs	Genome data mining and physiological and biochemical analysis, ‘dynamic degradation of TAG’: by controlling Sco6196, mobilize the TAG pool and increase polyketide biosynthesis	*Streptomyces*	[18]
Guaia-6,10 (14)-diene	FPP	Expression of synthetic gene FgJ02895, construction of a series of mutants to increase MVA and the downstream synthesis pathway	*E. coli*, *S. cerevisiae*	[19]
Hypoxanthine	5-phosphoribosylpyrophosphate, glutamine	Deregulation of PurR (Regulatory proteins), site directed mutation of key enzyme. Increase precursor accumulation and disrupt branch pathways	*E. coli, S. cerevisiae*	[109]
Artemisinic acid	Amorphadiene	Expresion of CYP71AV1, CPR1, CYB5, ADH1, ALDH1, regulate the MVA pathway, optimize the upstream and downstream and reduce the branch pathway	*S. cerevisiae*	[94]
Linalool	Mevalonic acid	Expression of synthetic gene LIS, LIS, overexpression of upstream gene tHMG1 to increase the MVA pathway	*S. cerevisiae*	[110]
Carnosic acid	GGPP	Overexpress tHMGR, knock out LPP1, MVA pathway optimization, increase the vitality of transcription factors, reduce the branch pathway	*S. cerevisiae*	[111]
Racemic naringenin, pinocembrin	4-coumaric acid,cinnamic acid	Selects a deletion of native pikromycin polyketide synthase gene strain, expression of synthetic gene CCL, CHS under the control of a single ermE promoter	*S.venezuelae*	[86]
Stilbene resveratrol, isorhapontigenin	4-coumaric acid, Caffeic acid	Expression of 4CL1,STS, optimization of precursor conversion and cyclization of the bulky ferulic acid precursor by metabolic engineering and protein engineering	*E. coli*	[85]
Taxol-5α-l	Glycerol	Expression of CYP725A4, tcCPR, optimize P450 expression, reductase partner interactions, N-terminal modifications	*E. coli*	[112]

## Data Availability

Not applicable.

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
