# Peer review of "Multi-Level Optimization and Strategies in Microbial Biotransformation of Nature Products"

_molecules, 2023, doi:10.3390/molecules28062619_

Round 1
Reviewer 1 Report
The paper ‘Multi-level Optimization and Strategies in Microbial Biotransformation of Nature Products’ by Dan et al review the optimization strategies of microbial biotransformation, including selecting potential strains, optimizing culture and fermentation conditions, synthetic biology strategies, combination chemical synthesis and biosynthesis, multi-strains co-culture strategy, and multi-omics strategies. This review provides a comprehensive summary of optimization strategies for microbial transformation.
Comments:
1. The term “transformed strain” used in this review is confusing. I suggest replace it with other words.
2. The subsection 3.1 in line 260 has the same title with subsection 3.2 in line 351.
3. Bioinformatics and genomics have proved the advantages in identifying the new clusters. I think bioinformatics and genomics could be discussed in section 2 “Selection of transformed strains”.
4. The section 6 “step-wise biotransformation strategy” mainly discussed the co-culture strategy used in biotransformation. I think this part could be incorporated into section 4 “Microbial transformation based on synthetic biology”.
5. There are some grammar errors should be checked, e.g., in line 535, “study” should be “studies”. In line 539-543, the sentence is too long and confusing, and should be split into several shorter sentences. Please check and correct any grammar errors in the review.
Author Response
Dear Reviewers,
Thanks a lot for reviewing our paper “Multi-level Optimization and Strategies in Microbial Biotransformation of Nature Products (Molecules-2250869)”. We appreciate your approval of our last manuscript revision. Your comments are all valuable and very helpful for revising and improving our paper, as well as the important guiding significance to our research. We have studied the comments carefully and have made revision which emphasis on the text in red font.
Now, we have modified our manuscript point by point according to the comments of reviewers. They mainly include:
Reviewer1:
1.Comment: The term “transformed strain” used in this review is confusing. I suggest replace it with other words.
Reply: Thank you for your correction. Through the review of relevant literature, we have modified the "transformation strain" to "microbial strains or transformed microbial strains". Please check the red font of the corresponding part of the text.
2.Comment: The subsection 3.1 in line 260 has the same title with subsection 3.2 in line 351.
Reply: Thank you very much for your reminder. We are sorry for such a small mistake. We have changed the titles of 3.1 and 3.2 to: “3.1. Solid fermentation and liquid fermentation”; “3.2. Optimization of fermentation conditions”.
3.Comment: Bioinformatics and genomics have proved the advantages in identifying the new clusters. I think bioinformatics and genomics could be discussed in section 2.
Reply: Thank you very much for your guidance. Under your suggestion, we have added the discussion of “2.4. Selection of microbial strains by bioinformation and genomics” in section 2 “Selection of transformed strains”. Please check the modifications in the newly uploaded "revised manuscript" file.
4.Comment: The section 6 “step-wise biotransformation strategy” mainly discussed the co-culture strategy used in biotransformation. I think this part could be incorporated into section 4 “Microbial transformation based on synthetic biology”.
Reply: Thank you very much for your guidance. Under your suggestion, we incorporated the contents of section 6 into the contents of section 4, and change the title of the original section 6 into “4.4 Multi-strain collaborative biotransformation strategy”. Please check all the above modifications in the newly uploaded "revised manuscript" file.
5.Comment: There are some grammar errors should be checked, e.g., in line 535, “study” should be “studies”. In line 539-543, the sentence is too long and confusing, and should be split into several shorter sentences. Please check and correct any grammar errors in the review.
Reply: Thank you for your carefully correction again. All the above minor errors and grammar errors have been modified in the corresponding parts of the text. Please check the red font of the corresponding part of the text.
Reviewer 2 Report
The authors presented the review on the multi-level optimization and strategies in microbial biotransformation of natural products. The content of this manuscript is well organized. This manuscript contains content that is of interest to experts in this field as well as non-experts. The manuscript has a merit to be published in Molecules. However, there are some suggestions which would improve the quality of the manuscript.
1. The relevance of the topics covered in this review to the work of the authors to date should be mentioned in the introduction section.
2. The authors need to mention in more detail whether a review article similar to the content of this manuscript was previously published.
3. Is the content covered in this manuscript comprehensive for all papers reported on this topic so far? Or does it only cover some representative reports? The authors should mention this point in the manuscript.
4. The authors should add structural formulas for all compounds covered in this manuscript. Also, for microbial biotransformation of natural products, the reaction scheme should be added.
5. The authors should add the structures of "substrate" and "obtained compounds" in table 1, and "products" and "precursor" in tables 2 and 3, respectively. Information on the structural formula would be very useful to readers of this journal.
Author Response
Dear Reviewers,
Thanks a lot for reviewing our paper “Multi-level Optimization and Strategies in Microbial Biotransformation of Nature Products (Molecules-2250869)”. We appreciate your approval of our last manuscript revision. Your comments are all valuable and very helpful for revising and improving our paper, as well as the important guiding significance to our research. We have studied the comments carefully and have made revision which emphasis on the text in red font.
Now, we have modified our manuscript point by point according to the comments of reviewers. They mainly include:
Reviewer2:
- Comment: The relevance of the topics covered in should be mentioned in the introduction section.
- Comment: The authors need to mention in more detail whether a review article similar to the content of this manuscript was previously published.
- Comment: Is the content covered in this manuscript comprehensive for all papers reported on this topic so far? Or does it only cover some representative reports? The authors should mention this point in the manuscript.
Reply: Thank you for your suggestions. With regard to the three questions you mentioned above, we have added the corresponding contents in the last paragraph of the introduction of the paper as follows: In the past 10 years, our research group has carried out a lot of research work on the biotransformation of plant natural products by endophytic fungi, and made a series of research progress. In order to make this research work better and provide reference for relevant researchers, we carried out this review. This review covers the optimization strategies of microbial biotransformation, including selecting potential strains, optimizing culture and fermentation conditions, synthetic biology strategies, combination chemical synthesis and biosynthesis, multi-strains co-culture strategy, and multi-omics strategies; 124 references from 1999 to 2022 are cited. In addition, literature review in relevant fields shows that this is the most comprehensive and systematic latest review of multi-level optimization and strategy of microbial biotransformation of natural products. In the past, the review of microbial transformation mainly focused on the presentation of a certain type of special substances, such as sterols, flavonoids, saponins, triterpenes, and so on, while ignoring the multiple strategies and optimization of biological biotransformation. Therefore, such a systematic summary has important reference and guiding significance for the research of microbial biotransformation of natural products.
- Comment: The authors should add structural formulas for all compounds covered in this manuscript. Also, for microbial biotransformation of natural products, the reaction scheme should be added.
Reply: Thank you very much for your guidance. This review focuses on optimization strategies of microbial biotransformation, including selecting potential strains, optimizing culture and fermentation conditions, synthetic biology strategies, combination chemical synthesis and biosynthesis, multi-strains co-culture strategy, and multi-omics strategies. Compared with the traditional biotransformation review, the biotransformation reaction of compounds is not main content to be discussed in this review, so we think it is unnecessary to add structure and reaction scheme of all the compounds. We have added the structural formula of representative compounds of Table 1-3, please check the corresponding modification in Supplementary materials.
5.Comment: The authors should add the structures of "substrate" and "obtained compounds" in table 1, and "products" and "precursor" in tables 2 and 3, respectively. Information on the structural formula would be very useful to readers of this journal.
Reply: Thank you very much for your guidance. We have added the structural formula of all the compounds in tables 1-3, please check the corresponding modification in Figure S1-S3.